# The Impact of the Nutritional Knowledge of Polish Students Living Outside the Family Home on Consumer Behavior and Food Waste

**DOI:** 10.3390/ijerph192013058

**Published:** 2022-10-11

**Authors:** Robert Nicewicz, Beata Bilska

**Affiliations:** 1Faculty of Human Nutrition, Warsaw University of Life Sciences (SGGW-WULS), Nowoursynowska 159C, 02-776 Warsaw, Poland; 2Department of Food Gastronomy and Food Hygiene, Institute of Human Nutrition Sciences, Warsaw University of Life Sciences (SGGW-WULS), Nowoursynowska 159C, 02-776 Warsaw, Poland

**Keywords:** food waste, students, nutritional knowledge, young people, best-before date

## Abstract

Food waste remains a major problem for the world and food security. Despite the fact that consumers are significant producers of food waste, little research attention has been paid to young people who are in college and living away from the family home. The present study aimed to assess food waste among college students living away from the family home, taking into account the nutritional knowledge acquired during college. In addition, the goal was to compare how nutritional knowledge affects food waste and consumer behavior in the study group. Descriptive statistics were performed on behaviors with food management at home, as well as shopping planning and self-shopping at the store, for the total respondents, students with nutritional knowledge and students without such knowledge. A chi-square test was performed to show whether the field of study influences the frequency of food throwing away and the appropriate management of excess food. Spearman’s rank correlations were calculated for the frequency of food discarding, the frequency and amount of shopping and the field of study. The results indicate that the field of study determined food discarding and appropriate food management (*p* < 0.05), while it was found that fermented dairy beverages, fruits and vegetables and bread were the most frequently discarded foods in both study groups (*p* < 0.05). Students with nutritional knowledge were less likely to throw away food compared to students without nutritional knowledge and were better at managing excess food.

## 1. Introduction

Food waste is a global problem involving every distribution channel in highly developed countries. According to [1], “food waste refers to food fit for human consumption that is discarded, whether or not it is stored beyond its expiry date or left to spoil. This is often due to food going bad, but can also be due to other reasons such as oversupply in markets or individual consumer shopping/eating habits.” The Food and Agriculture Organization of the United Nations reports that 1.3 billion tons of food that are still fit for consumption are wasted worldwide each year, accounting for one-third of our global production [2]. The FUSIONS project was established in the European Union, which estimated that nearly 100 million tons of food waste are generated annually [3]. The average recorded in Poland is 247 kg/per capita/year, which places this country in 5th place [4]. Just ahead are the United Kingdom, Germany, France, and the Netherlands [5], while the lowest food waste was noted in Malta, Romania, and Slovenia [4].

In cooperation with two research teams of the Institute of Environmental Protection and Warsaw University of Life Science within the PROM project, almost 5 million tons of food are wasted in Poland [6]. It identifies six primary aspects that contribute to food waste: transport, catering, trade, agricultural production, processing and households. The latter is the cause of the greatest losses because as much as 60% of thrown-away food comes from this area. One of the UN Sustainable Development Goals was to cut food waste in half [7]. Food waste is also associated with excessive energy and water use, which are essential for production [8,9]. Reducing food waste contributes to greater savings [10], achieving a sustainable food system [11,12,13], and environmental improvements [14,15,16,17,18]. Study [19] indicated that food waste is determined by demographic, socioeconomic, cultural, psychological, and behavioral factors.

The most common causes of food waste in households include exceeding expiration dates, excessive shopping, oversized portions of prepared foods, and food spoilage [20,21,22,23,24]. Numerous studies show that the most frequently thrown away food products are bread, fruits and vegetables, processed meats, dairy products, and prepared meals [22,23,25,26,27]. According to [28], consumers have a negative attitude toward food waste, but data of [2,3,6] show that consumers waste a lot of food.

The COVID-19 pandemic has affected consumer shopping habits [23,29,30]. They were more likely to shop on their own or give a shopping list to a loved one. Despite planning their purchases and making a list, exceeding the expiration date was the main reason for throwing away food. On the positive side, there was a reduction in the amount of discarded food products [23,30,31,32,33,34]. More consumers have begun to use excess food for freezing, further processing, or animals [35].

### 1.1. Food Waste among Students/Young People

The fact that young consumers waste more food compared to older people was proven by numerous studies [21,36,37,38,39]; hence, the interest in this group of respondents in our study. Over 1.2 million students studied in Poland in the 2020/2021 academic year [40]. Some students are concerned about food waste’s economic and environmental impacts, but many ignore the problem [41]. Clark and Manning [42] argue that young adults and college students are aware of the effects of food waste and try to prevent it. However, Islam [43] indicates that young people are wasting food at a massive rate, but on the other hand, the zero-waste movement is also visible [44,45]. Numerous studies have shown that college students are more likely to throw away food and less likely to eat leftovers and consume food supplies [12,21,39,46,47]. A total of 71% of students from Europe, Asia, and Africa waste food [38], while more than 90% of Polish students admit to food waste, throwing away the food on average once a week [36]. Among students, the most common reasons for throwing away food were exceeding the expiration date, storing food for too long, not having enough time to use up stored ingredients, and preparing too-large portions of meals [36,38]. Among a review of school and college cafeterias and through teacher observations, large-scale food discarding among students has been noted [48,49,50]. Among the food waste from students’ plates leftover cereal products and vegetables were the most frequent [51].

### 1.2. Education’s Influence on Food Waste

Food-waste education resulted in food discarding [49,52,53,54]. Education on sustainable food systems and general food topics can help prevent food waste among youth through better food choices [55,56,57,58,59]. Introducing education among young people about sustainable and healthy meals can contribute to long-term dietary change that will reduce climate impacts and improve population health [60]. In studies among parents, children, and adolescents, there is a significant association between knowledge about food safety, food planning and management, and avoidance of throwing away food [61,62]. Articles conducted among young people with nutritional knowledge do not appear frequently, except [62]. However, one can find studies conducted in natural-science universities [36,39]. A study by [63] found that students’ knowledge and awareness do not at all go hand in hand with reducing food waste. Hence, it is important to conduct further research and analysis. Ko and Lu [64], on the other hand, said that education in the human-nutrition and food-service professions helps manage leftover food. Students with an increased interest in food and food waste are more likely to avoid throwing food away [65].

### 1.3. Strategies to Reduce Food Waste

According to the 12th Sustainable Development Goal ‘Responsible Consumption and Production’, we are to reduce food waste in the supply chain by as much as 50% by 2030 [7]. To this end, numerous solutions are proposed to meet these goals. As a result of these provisions, the following solutions are proposed: changes in food packaging; reducing portion sizes; standardizing labeling dates; selling uglier, less-perfect products with shorter expiration dates in stores; using uneaten food for animals; implementing community composting and operating community gardens; educational campaigns on food waste; and taxing food waste [7,66,67]. As the most common causes of food waste are behavioral factors and problems with shopping planning, food storage and preparation, there should be an emphasis on education campaigns and intervention programs focusing on these factors [68]. Moraes et al. [69], on the other hand, showed methods such as education and awareness, policy, donations, research, and storage, demand control, and logistics. Education is an important component, but it depends on audience engagement [70,71,72]. Any legal changes should serve to address food waste [73]. Policy practices to prevent food waste are aimed at not throwing food away despite their imperfections [69]. Governments should enforce this law [72]. Scientific research is expected to help understand consumer behavior and the impact on food discarding in every food distribution chain, including food service and hospitality [73]. This is helpful for further educational and legal action. Consumer research methods are often based on consumers’ attitudes, meal planning, shopping lists, etc. [69]. In order to reduce food waste, it is necessary that all participants in the food and distribution chain participate, which will help comprehensively and logistically achieve one of the sustainable development goals [74,75].

### 1.4. Objective

Therefore, the main objective of the research presented in this study was to evaluate the food-wasting behavior of students living outside the family home, with particular emphasis on nutritional knowledge. In addition, included were the sub-goals: to find out the differences in shopping behavior and household activities of students with nutritional knowledge and students of other majors, to compare the influence of nutritional knowledge on food wasting among students, and the frequency of throwing away given food items and dealing with surplus food. Attention was also paid to aspects related to the proper distinction of date labels on food products in the group of students with and without nutritional knowledge and correlations between frequency and volume of grocery shopping and food discarding.

The scope of the study included: analysis of Polish and foreign literature and research reports on food wastage, development of the author’s research tool to find out the influence of students’ nutritional knowledge on food wastage, conducting a quantitative survey among students living outside their family home, and collecting data and their statistical analysis.

We hypothesize that: students with nutritional knowledge are less likely to waste food and better manage food in their households compared to students without such knowledge.

The present study extends previous research conducted in Poland [39,63], which considered food waste among students. An additional advantage of this work is that it compares respondents who have knowledge of food and nutrition to those who do not have such knowledge.

This paper is organized into five sections. The research material and methodology are described in Section 2. Section 3 reveals study findings, which are then discussed according to the body of literature in Section 4. Finally, Section 5 contains the conclusions.

## 2. Materials and Methods

### 2.1. Data-Collection Process

The cross-sectional survey was conducted between October 2021 and May 2022. CAWI (computer-assisted web interview) technique was used to collect data [76,77]. Confidentiality of the data was ensured as well as anonymity ensured. This method was an easy and quick way to obtain responses from this age group because they spend the most time using the internet. A pilot study was conducted among 100 students, confirming the questionnaire’s validity.

The sample was recruited from among students based on a social-media advertisement. The ad included a link to a questionnaire with a request for those who met the inclusion criteria to complete the questionnaire. The following inclusion criteria were considered: students, studying in Poland, age 18–26, student status, and gave informed consent to participate in the study. The country has 1.2 million college students [40]. The survey included 1208 respondents between the ages of 18 and 26 who were university students, accounting for 0.1% of all Polish students. A total of 45% of the respondents were students with knowledge of nutrition (N), and 55% were students in other fields (O). Students majoring in human nutrition, food technology, dietetics, catering and hospitality were considered to have knowledge of nutrition. The remainder were counted as non-nutrition majors.

### 2.2. Description of Questionnaire

The study was conducted using a proprietary questionnaire developed on the basis of previous literature [22,24,62,78]. The questionnaire was divided into two parts (Figure 1). In the first part, students answered questions about the frequency and size of their purchases. Students were also asked about throwing food in the trash. Those who answered affirmatively had to answer questions about why food is wasted and how often they throw away the items listed below. Later questions were common to both groups of respondents (food wasters and non-wasters). Students were also asked if they knew the difference between labelling with an expiration date and a use-by date. Those who answered affirmatively were asked to define the term “best before”. All respondents were also asked what they thought the different food labels were. A reliability test was conducted—Cronbach’s alpha coefficient. A result of α = 0.81 was obtained, indicating that the test is reliable.

### 2.3. Statistical Analysis

The collected empirical data were statistically analyzed using Statistica 13.3 (Figure 2). The mean, median, confidence interval (CI), standard deviation (SD) and kurtosis were calculated [79,80] for household chores and shopping behavior by the general group, students with nutrition knowledge and students from other majors. A chi-square test was used [24,29] to determine the effect of nutrition knowledge compared on performing household activities and shopping behavior with students from other majors. The Mann–Whitney U test was used [29,30,81] to compare the two study groups in terms of home and grocery-store activities and frequency of discarded food items. In both study groups, Spearman’s rank correlation coefficient [29,82] was calculated between frequency and amount of shopping and frequency of discarded food items among students with nutrition knowledge, as well as students from other majors.

## 3. Results

### 3.1. Characteristics of Respondents

A total of 1208 students participated in the survey. A larger percentage of respondents were female (Table 1). Students between the ages of 18 and 26 participated in the study. More than 50% were students studying in fields other than nutrition. The others were studying nutrition-related courses. Students mostly lived in rented apartments, and rooms. The least of them stayed in dormitories.

### 3.2. Shopping Behaviors and Home and Store Activities

Students surveyed most often store 2–3 times per week (59.6%), while one in five attends the store 4–5 times per week. Slightly fewer respondents said they shop weekly or less often, and only a few said they shop daily (5.3%). Respondents were most likely to shop at supermarkets and discount stores (69.5%). Other responses included: neighborhood stores (17.9%) and hypermarkets (10.6%). Only a few individuals did their shopping online. Responses regarding the size of purchases varied quite a bit; however, one-third of students indicated that they buy between 6 and 10 food items. The remaining responses ranked as follows: 11 to 15 food items (25.2%), 1 to 5 food items (23.8%), more than 20 food items (9.3%), and 16 to 20 food items (7.9%).

Students were most likely to control the number of food items at home, the expiration date on the package, put away items with longer expiration dates and put away items with shorter expiration dates (Table 2). There was a statistically significant difference between the indications of students with nutrition knowledge and students with non-nutrition majors (*p* < 0.05), except for controlling food items at home. Students with nutritional knowledge were slightly more likely to prepare a shopping list compared to students without such knowledge. However, they all generally try to stick to the list when shopping in the store. Students with nutrition knowledge are less likely to browse the assortment of products on the store shelves and rarely take the products on the list and the ones they want. A good habit among the general student population tends to be checking expiration dates. Students who are knowledgeable about nutrition usually always do this. As for shopping for “stock,” they were characterized by responses of “rarely” and “never”.

### 3.3. Food Waste among Students

Among students with nutrition knowledge, less than 30% throw away food, while 85% of students from other majors admit to wasting food (*p* < 0.05). All students with nutrition knowledge threw away food due to shelf life (n = 160) (Table 3). However, this is significantly fewer students than students from other majors (n = 528). Among students with nutrition knowledge, “shopping too much” (n = 40), “improper storage” (n = 96), and “preparing too large portions of meals” (n = 104) were also more frequently indicated. In comparison, the responses among students without nutrition knowledge were (n = 80; n = 304; n = 295), respectively. For the remaining reasons, response rates were higher among students without nutrition knowledge.

All students who contributed to food waste threw away a variety of food items (Table 4). The most commonly discarded foods were bread, fruits, vegetables, prepared meals, and fermented milk drinks. Students from non-nutrition majors were more likely to throw away foods in the trash than students with nutrition knowledge, except for meat preparations, white cheese, fermented milk drinks, and fruits and vegetables (*p* < 0.05). A higher percentage of respondents studying nutrition appeared with the response “once a month or less often”; however, the above exceptions indicate that among the 160-person wasting group, nutrition majors also happen to throw away produce, with greater frequency, such as bread.

Calculated correlations of Spearman coefficients showed that both groups of students (with and without nutritional knowledge) report an increased frequency of throwing away food products from the three main groups, i.e., bread, fresh fruits and vegetables, and fermented milk drinks (*p* < 0.05) (Table 5). Weak (0.2 < r < 0.4), moderate (0.4 < r < 0.7) and fairly strong correlation (0.7 < r < 0.9) relationships between frequency and volume of shopping and throwing away of these food products are evident here. In addition to these products, a moderate relationship (0.4 < r < 0.7) with ready-to-eat meals and milk (bottle and UHT sterilized) and a weak correlation (0.2 < r < 0.4) but significant relationships with meat, poultry and cold meats were also identified among students with no nutritional knowledge.

### 3.4. Managing Food

The most common handling of excess food among students was freezing and preparing another dish (*p* < 0.05) (Table 6). Despite having nutritional knowledge, students with nutritional knowledge were less likely to pasteurize food (n = 144), compared to students without nutritional knowledge (n = 560). Preparing other dishes from food that was in excess is performed by more students without nutrition knowledge (n = 592), compared to students with nutrition knowledge (n = 473). For those in need, more students from nutrition majors (n = 81) donate excess food than non-nutrition majors (n = 72). However, the most important aspect is food waste among students. The vast majority of students without nutrition knowledge (n = 568) compared to students with nutrition knowledge (n = 160), waste food.

### 3.5. Differences in Date Marking on Food Products

Students with nutrition knowledge indicated mostly correct answers regarding products having a best-before date (Table 7). Indications ranged from about 70% for rennet cheese to as high as 95% for cereal products such as pasta, groats, and flours, while students with non-nutrition majors indicated correct answers between 31% for rennet cheese to almost 88% for pasta, groats, and flours (*p* < 0.05).

## 4. Discussion

Our study shows that students mostly shop 2–3 times a week and attend supermarkets and discount stores. Ginea and Ghiuta [37] also indicate that young people are more likely to shop at supermarkets, but there is a higher proportion of those who shop daily. Some young people check the fridge and cupboards and make a shopping list before going out [21,36,39]. In our results, it was proven that students most often perform such activities. Young respondents who planned their shopping tended to avoid throwing away food [83]. Several studies also indicate that young people lack knowledge about food storage and do not make a shopping list [42,84,85], which is noticeable among students without nutritional knowledge. When shopping, students check the expiration date on the package, but they also often buy on impulse [39]. It also happened that young people were encouraged by attractive promotions to buy more food [37]. Many students with nutritional knowledge are quite susceptible to all kinds of promotions prompting them to buy more, but it has been shown that they make sensible purchases and follow a list when making them [63].

Young people, including students, often eat out or order take-out food, which can lead to food waste in their households [13,86]. It is difficult for young people to estimate how much produce they need in their homes and how much food needs to be cooked [87,88]. However, leaving plate leftovers at eating establishments also contributes to food waste [89]. Lack of food and nutrition knowledge and cooking experience was also a reason [87,90]. This is also indicated by our study, which shows that students with nutritional knowledge manage food better and waste less. Food waste among natural-science students (including those with nutrition knowledge) in Poland and Sweden is very high [36].

On the other hand, at the University of Life Sciences in Poznań, significantly more food waste is observed among students than employees [39]. However, age is not a differentiating factor in food waste [22]; however, Karunasena [91] argued that there are intergenerational differences between the behaviors of young consumers and older adults. It has been noted that college students throw away food more often than college employees [39]. Many young people tend to pay attention to the expiration date on the package [37]; however, as our study showed, they throw away a lot of food, especially people without nutritional knowledge. It is estimated that young people throw away between 250 and 2000 grams of food waste in the trash [30,34]. Among the reasons for food waste among young consumers and students (including those with nutritional knowledge) are exceeding the best-before date, storing for too long, poor product quality, and preparing too large portions of meals [21,36,37,39,63]. These data collated with our results. The most frequently discarded foods were bread, fruits and vegetables, and foods of animal origin [39,63]. Despite elementary knowledge about nutrition and food in some students, these behaviors also occurred [63], which also agreed with our results obtained in this group of respondents. Burlea-Schiopoiu [34] showed that the COVID-19 pandemic had a positive impact on young people’s food-wasting behaviors. Among the general population, the percentage of respondents who throw food in the trash decreased [23].

As shown in our study, a large percentage of respondents “prepare” meals from excess food and leftovers. This is also confirmed by authors of other studies [83,92,93,94,95]. Ghinea and Ghiuta [37] showed that some respondents use excess food for animals, but about a third of them throw away food. Many young consumers are afraid to consume longer stored food, so they prefer to throw it away [85]. About two-thirds of Polish university students (also with nutritional knowledge) indicated that they allocate surplus food for further processing [36] and manage food well in their own households [63].

In our study, students with nutritional knowledge were more aware of the differences in date labels and which products have a best-before date. Zabłocka et al. [36], in a study of natural science college students including those with nutritional knowledge, also indicated a high percentage of familiarity with date labels. Their behavior was also correlated with good knowledge of food storage [64]. Broshius’ [96] study found some discrepancies in respondents’ answers about expiration dates and food storage. In some cases, false and inconsistent statements were given. However, ref. [83] Marek-Andrzejewska and Wielicka-Regulska indicated that students’ ignorance of how to label dates (as in people without nutritional knowledge) is associated with ignorance of the differences between dates, not that products with expiration dates can be consumed after the expiration date. This is also confirmed by [97], that consumers do not understand the difference between “use by” and “best before” labels. Although products often show no signs of spoilage, they are discarded, such as eggs [37]. This may be because they need to be stored in the refrigerator [98]. A study on consumer perceptions of products with expiration dates indicated that young consumers were reluctant to buy products with short expiration dates [98] and the expiration date on the package is important to them [99]. They were more willing to check the dates of less durable products such as meat, fish, milk and milk products. Thus, expiration date plays an important role in consumer behavior when choosing food products [99,100,101].

### Implications for Research and Practice

This study shows that students waste a lot of food and it is important for young people to combat food waste. It is this population that should care about the environment they will live in for decades to come. This study strengthens and extends the literature on food waste by conducting analyses among college students who no longer live with their parents and have to support themselves and make decisions on their own, including those related to food. This brought some novelty to this type of research as it has typically focused on the general student population, meaning those who live with family as well. In addition, comparing students with nutritional knowledge to students from other majors is novel, because there are not many studies on such comparisons.

Food waste is a significant global problem and young people need to be activated to change their behavior. This study can guide educational efforts and contribute to further research in this area. Young people are seen as a generation that does not care about food. Our survey can help develop an educational campaign about reducing food waste and managing food properly. Therefore, education among students regarding food wastage and its consequences should be considered [59,69,97,102,103,104]. They should have a broader picture of the consequences associated with food wastage and they should be made aware that wastage itself leads to serious economic, environmental, moral, and social consequences. Direction should also be given to these people regarding proper food management. Especially those who live outside the family home should take care of their budget and manage food and finances wisely. For example, it is worth introducing awareness to young people through apps or social media. This path is most suitable for the generation that uses phones, computers, and other mobile devices the most. Such apps would help them plan and manage their finances and food and allow them to control their behavior. They could also give practical tips on how to handle particular foods.

Authors should discuss the results and how they can be interpreted from the perspective of previous studies and of the working hypotheses. The findings and their implications should be discussed in the broadest context possible. Future research directions may also be highlighted.

## 5. Conclusions

The quantitative study showed that nutrition students are less likely to buy products they crave, more likely to prepare shopping lists, and more likely to notice and reach for novelties to try to incorporate them into their diets in the future. For this reason, unfortunately, despite preparing a shopping list, they happen to reach for products outside of it. Those who study nutrition and derivatives are less likely to throw away food for knowledge-related reasons, such as expiration dates, shopping too much, or poor-quality products. Excess food in students’ households is partially allocated to animals and this does not depend on the field of study and knowledge, while managing excess food through processes, such as pasteurization, curing or freezing, is more common among students with nutrition knowledge gained in college. Nutrition knowledge influences the frequency of discarding products from different food groups; however, frequent discarding of fermented dairy beverages, bread, and fruits and vegetables is also noted among this group of students. Consumers’ knowledge and awareness do not fully reflect the activities aimed at reducing food waste, which was also demonstrated in our study.

As a result of the analyses, it is possible to confirm the hypothesis that students with nutritional knowledge are less likely to waste food and better manage food in their households compared to students without such knowledge, despite the noticeable errors in food wasting or management demonstrated during the analyses.

This study can guide educational efforts and contribute to further research in the area of food waste. Our study may help in developing an educational campaign to reduce food waste and ensure proper food management. Therefore, educating students about food waste and its consequences should be considered. They will provide practical tips on how to handle certain food items and give a good foundation for action on the study population through these education and training programs.

### Limitation of the Study

Despite the large sample size of 1208 students, it should be remembered that the sample selection was voluntary, so we cannot speak of its representativeness. It would be worth repeating the survey among Polish students, but also among students from other countries (e.g., the European Union), because inferring in the group of Polish students does not prove that students from other countries also exhibit the same food-wasting behavior. This could be performed using, for example, a diary to measure the amount of food wasted.

## Figures and Tables

**Figure 1 ijerph-19-13058-f001:**
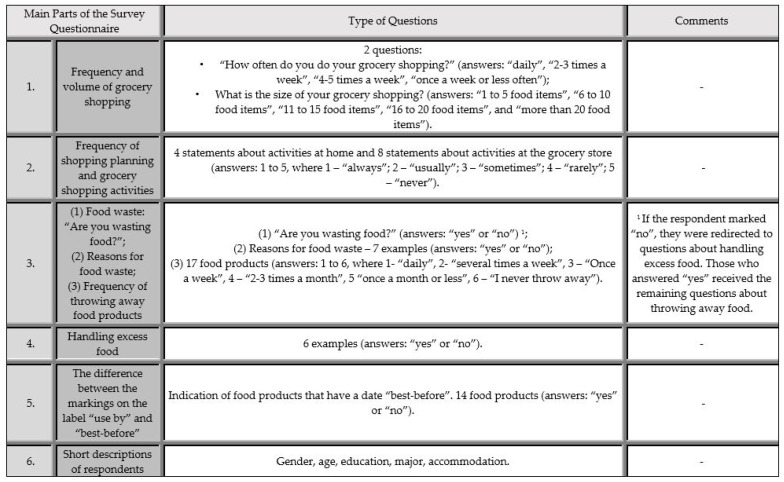
Division of the questionnaire.

**Figure 2 ijerph-19-13058-f002:**
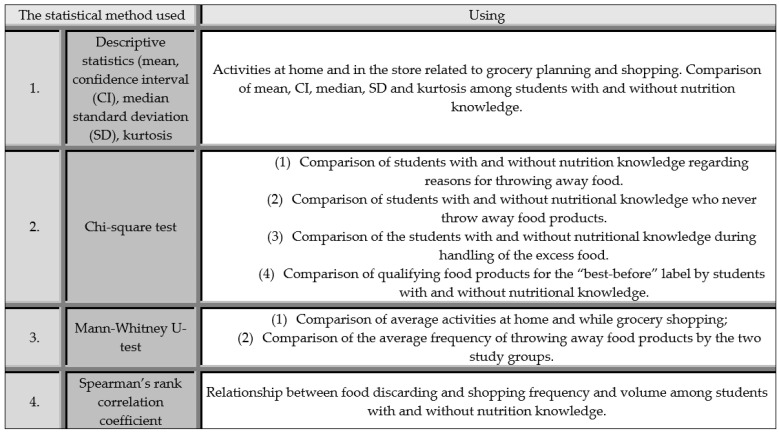
The statistical method used.

**Table 1 ijerph-19-13058-t001:** Characteristics of the study sample.

KERRYPNX	Total	
	Number of Respondents (n)	Percentage (%)
Total	1208	100.0
Gender		
Female	760	62.9
Male	448	37.1
Age		
18–22	600	49.7
23–26	608	50.3
Study major		
Nutrition and related	544	45.0
Other	664	55.0
Current residence		
Dormitory	152	12.6
Rented room	416	34.4
Rented apartment	640	53.0

Source: Own survey.

**Table 2 ijerph-19-13058-t002:** Mean, confidence interval (CI), median, standard deviation (SD), and kurtosis for shopping behavior and home and store activities.

	Activities	Descriptive Statistics															
		Mean			*p*-Value *	95% CI			Median			SD			Kurtosis		
		All ^1^	N ^2^	O ^3^		All ^1^	N ^2^	O ^3^	All ^1^	N ^2^	O ^3^	All ^1^	N ^2^	O ^3^	All ^1^	N ^2^	O ^3^
Activities at home	I control the amount of food I have at home	1.69	1.63	1.73	b.d	1.65–1.73	1.58–1.69	1.68–1.69	2.00	2.00	2.00	0.73	0.66	0.78	2.35	0.92	2.66
	I check the expiration date of products in my home	1.95	1.69	2.16	<0.05	1.89–2.00	1.63–1.75	2.08–2.24	2.00	2.00	2.00	0.95	0.73	1.06	0.51	1.03	−0.22
	I put products with longer expiration dates behind products with shorter expiration dates	2.44	1.99	2.82	<0.05	2.37–2.52	1.91–2.06	2.71–2.93	2.00	2.00	2.00	1.33	0.90	1.50	−0.64	0.74	−1.38
	I prepare a list of products needed	1.90	1.72	2.05	<0.05	1.84–1.96	1.64–1.80	1.97–2.13	2.00	1.00	2.00	1.01	0.92	1.05	0.42	1.62	−0.11
Store activities	I buy products according to my shopping list	1.91	1.69	2.08	<0.05	1.86–1.96	1.62–1.76	2.01–2.15	2.00	2.00	2.00	0.89	0.81	0.92	2.23	3.66	1.77
	Only when grocery shopping do I browse the assortment of products on the store shelves	3.13	3.53	2.80	<0.05	3.07–3.19	3.44–3.62	2.72–2.87	3.00	4.00	3.00	1.08	1.05	0.98	−0.54	−0.26	−0.67
	I happen to take a product off my shopping list	2.76	3.43	2.22	<0.05	2.70–2.82	3.34–3.51	2.16–2.28	3.00	4.00	3.00	1.10	1.05	0.81	−0.75	0.08	−0.77
	I choose the foods I want to eat	2.76	3.49	2.17	<0.05	2.70–2.83	3.40–3.57	2.11–2.23	3.00	4.00	3.00	1.13	1.01	0.83	−0.79	−0.08	0.52
	I often reach for products offered as a “novelty” to try them	3.60	3.79	3.43	<0.05	3.54–3.65	3.72–3.87	3.36–3.51	4.00	4.00	4.00	0.96	0.88	0.98	0.45	1.46	0.07
	I check the use-by date before I put a food item in my shopping cart	1.91	1.54	2.20	<0.05	1.85–1.97	1.47–1.62	2.12–2.29	2.00	1.00	2.00	1.09	0.88	1.15	0.10	1.33	−0.44
	I buy products with a short use-by date if they are discounted	2.89	2.78	2.98	<0.05	2.83–2.95	2.70–2.86	2.88–3.07	3.00	3.00	3.00	1.08	0.91	1.19	−0.61	−0.27	−0.87
	I shop “in stock”	3.66	4.35	3.08	<0.05	3.59–3.72	4.28–4.43	3.07–3.17	4.00	5.00	4.00	1.21	0.92	1.12	−0.71	−0.12	−0.61

^1^ all study participants (n = 1208); ^2^ students with nutritional knowledge (n = 544); ^3^ students from other majors (n = 664); 1—Always; 2—Usually; 3—Sometimes; 4—Rarely; 5—Never; * Mann–Whitney U test performed between the N mean and the O mean; Source: own survey.

**Table 3 ijerph-19-13058-t003:** Causes of food waste among students.

Cause	% of Respondents (n = 728)				*p*-Value *
	Yes		No		
	N ^1^	O ^2^	N ^1^	O ^2^	
Exceeding the use-by date	100.0	92.9	-	7.0	<0.05
Shopping too much	25.0	14.1	75.0	85.9	<0.05
Unpalatable product	55.0	73.2	45.0	26.8	<0.05
Poor quality product	35.0	56.3	65.0	43.7	<0.05
Improper storage	60.0	53.5	40.0	46.5	<0.05
Preparing too large a portion of meals	65.0	52.1	35.0	47.9	b.d
Lack of ideas and creative invention to use products of dishes	5.0	21.1	95.0	78.9	<0.05
Too much food received from parents	25.0	31.0	75.0	69.0	b.d

^1^ N—students with nutritional knowledge (n = 160); ^2^ O—students from other majors (n = 568); * Chi-square test; Source: own survey.

**Table 4 ijerph-19-13058-t004:** Frequency of food discarding among students.

Food Product	% of Respondents (n = 728)										Mean (1–5)		*p*-Value *	% of Respondents (n = 728)		*p*-Value **
	Daily (1)		Several Times a Week (2)		Once a Week (3)		2–3 Times a Month (4)		Once a Month or Less (5)					I Never Throw Away (6)		
	N ^1^	O ^2^	N ^1^	O ^2^	N ^1^	O ^2^	N ^1^	O ^2^	N ^1^	O ^2^				N ^1^	O ^2^	
Meat	-	-	-	1.4	5.0	5.6	5.0	11.3	50.0	47.9	4.56	4.55	<0.05	40.0	33.8	<0.05
Poultry	-	-	-	-	5.0	-	5.0	19.7	60.0	50.7	4.41	4.33		30.0	22.5	
Cold meats	-	1.4	-	5.6	-	2.8	10.0	14.1	45.0	45.1	4.43	4.46		40.0	31.0	
Cottage cheese	-	-	-	4.2	-	1.4	20.0	25.4	55.0	52.1	4.24	4.28		25.0	16.9	
Rennet cheese	-	4.2	-	1.4	-	4.2	10.0	12.7	65.0	69.0	4.29	4.23		20.0	8.5	
Fish	-	-	-	1.4	5.0	7.0	5.0	4.2	65.0	47.9	4.74	4.74		25.0	39.4	
Eggs	-	-	-	1.4	10.0	7.0	-	9.9	80.0	66.2	4.61	4.50		10.0	15.5	
Milk (bottle)	-	-	-	1.4	5.0	4.2	20.0	16.9	35.0	49.3	4.42	4.32		40.0	28.2	
Milk (UHT sterilized)	-	-	-	1.4	5.0	7.0	5.0	11.3	65.0	53.2	4.50	4.47		25.0	26.8	
Fermented milk drinks	-	-	-	1.4	20.0	15.5	15.0	35.2	60.0	43.7	3.90	3.95		5.0	4.2	
Fruits and vegetables	-	-	-	8.4	40.0	21.1	40.0	45.1	20.0	25.4	3.55	3.63		-	-	
Bread	5.0	2.8	-	8.4	25.0	26.8	45.0	35.2	25.0	26.8	3.79	3.64		-	-	
Pasta, groats, flours	-	-	-	5.6	-	4.2	5.0	2.8	25.0	7.0	4.57	4.47		90.0	80.3	
Dried spices and herbs	-	1.4	-	5.6	-	1.4	-	4.2	10.0	7.0	4.61	4.50		90.0	80.3	
Tea and coffee	-	1.4	-	5.6	-	1.4	-	2.8	10.0	8.4	4.64	4.54		90.0	80.3	
Sweets	-	-	-	1.4	-	4.2	5.0	5.6	10.0	4.2	4.70	4.63		85.0	84.5	
Ready-to-eat meals	-	1.4	-	2.8	15.0	5.6	30.0	35.2	40.0	49.3	4.07	4.00		15.0	5.6	

^1^ N—students with nutritional knowledge (n = 160); ^2^ O—students from other majors (n = 568); Mean^1–5^ calculated for “Once a month or less”, “2–3 times a month”, “Once a week”, “Several times a week”, and “Daily” responses; * Mann–Whitney U test performed between the N mean and the O mean; ** Chi-square test; Source: own survey.

**Table 5 ijerph-19-13058-t005:** Calculated coefficients of the Spearman rank between the shopping frequency and volume of purchases and the frequency of throwing away food product.

Food Product	Spearman’s Rank Correlation Coefficient			
	N ^1^		O ^2^	
	Shopping Frequency	Volume of Purchases	Shopping Frequency	Volume of Purchases
Meat	0.205	0.130	0.207	0.246 *
Poultry	0.108	0.173	0.195	0.178 *
Cold meats	0.157	0.195	0.202	0.188 *
Cottage cheese	0.234	0.183	0.247	0.191
Rennet cheese	0.104	0.171	0.124	0.179
Fish	0.096	0.124	0.149	0.131
Eggs	0.087	0.080	0.138	0.097
Milk (bottle)	0.345	0.173	0.238 *	0.171
Milk (UHT sterilized)	0.201	0.167	0.357 *	0.169
Fermented milk drinks	0.544 *	0.602 *	0.665 *	0.704 *
Fruits and vegetables	0.278 *	0.744 *	0.776 *	0.302 *
Bread	0.698 *	0.432 *	0.623 *	0.453 *
Pasta, groats, flours	0.088	0.060	0.091	0.087
Dried spices and herbs	0.075	0.045	0.089	0.093
Tea and coffee	0.075	0.045	0.089	0.093
Sweets	0.035	0.027	0.078	0.069
Ready-to-eat meals	0.075	0.045	0.423 *	0.543 *

* significance level *p* < 0.05; ^1^ N—students with nutritional knowledge (n = 160); ^2^ O—students from other majors (n = 568); Source: own survey.

**Table 6 ijerph-19-13058-t006:** Handling of the excess food by students.

Cause	% of Respondents (n = 1208)				*p*-Value *
	Yes		No		
	N ^1^	O ^2^	N ^1^	O ^2^	
I freeze	98.5	89.2	1.5	10.8	<0.05
I pasteurize/weave	26.5	84.3	73.5	15.7	<0.05
I prepare other dishes with them	85.3	89.2	14.7	10.8	<0.05
For animals	39.7	37.4	60.3	62.5	b.d
I give to people in need	14.7	10.8	85.3	89.2	<0.05
Throw away	29.4	85.6	70.6	14.4	<0.05

^1^ N—students with nutritional knowledge (n = 544); ^2^ O—students from other majors (n = 664); * Chi-square test; Source: own survey.

**Table 7 ijerph-19-13058-t007:** Qualifying food products for date marking methods: “best-before”.

Cause	% of Respondents (n = 1208)				*p*-Value *
	Yes		No		
	N ^1^	O ^2^	N ^1^	O ^2^	
Bread (packaged) **	82.4	59.0	17.6	41.0	<0.05
Pasta, groats, flours **	95.6	87.9	4.4	12.1	
Meat	5.9	20.5	94.1	79.5	
Fish	5.9	21.7	94.1	78.3	
Rennet cheese **	70.6	31.3	29.4	68.7	
Cottage cheese	10.3	22.9	89.7	77.1	
Fermented milk drinks	8.8	28.9	91.2	71.1	
Eggs **	92.7	68.7	7.3	31.3	
Milk (bottle)	10.9	29.3	89.1	70.7	
Milk (UHT-sterilized) **	75.0	39.8	25.0	60.2	
Dried spices and herbs **	89.7	79.5	10.3	20.5	
Tea and coffee **	77.9	49.4	22.1	50.6	
Sweets **	93.7	91.4	6.3	8.6	
Honey **	89.7	81.9	10.3	18.1	

^1^ N—students with nutritional knowledge (n = 544); ^2^ O—students from other majors (n = 664); * Chi-square test; ** products with the designation “best-before”; Source: own survey.

## Data Availability

Not applicable.

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
