# Peer review of "The Impact of the Nutritional Knowledge of Polish Students Living Outside the Family Home on Consumer Behavior and Food Waste"

_ijerph, 2022, doi:10.3390/ijerph192013058_

Round 1
Reviewer 1 Report
Dear Author(s),
Thank you for the opportunity to review this research dealing with consumer behavior and food waste.
However, although the issue represents a challenge under many perspectives, the paper presents some lacks. I have some minor concerns which I have to address you before I can suggest the submission of the manuscript to the “International Journal of Environmental Research and Public Health”.
I warmly encourage Author(s) to end the introduction introducing the structure of their manuscript.
What about the COVID-19 pandemic? Have you considered it? How has it influenced the opinion of respondents?
I suggest you to introduce the topic of ugly food and of strategies which can be pursued for reducing food waste also with reference to published case studies of other countries. As your manuscript presents only Polish suggestions, this would be interesting.
The following papers can be useful for the above-mentioned purpose:
· Fan, L., Ellison, B., & Wilson, N. L. (2022). What Food waste solutions do people support? Journal of Cleaner Production, 330, 129907.
· Varese, E., Cesarani, M. C., & Wojnarowska, M. (2022). Consumers' perception of suboptimal food: strategies to reduce food waste. British Food Journal.
· Ferro, C., Ares, G., Aschemann-Witzel, J., Curutchet, M. R., & Giménez, A. (2022). “I don't throw away food, unless I see that it's not fit for consumption”: An in-depth exploration of household food waste in Uruguay. Food Research International, 151, 110861.
I don’t feel qualified to judge about the English language and style.
Please accept my most sincere appreciation for your contribution.
Good luck!
Author Response
Thank you for your positive review.
I warmly encourage Author(s) to end the introduction introducing the structure of their manuscript.
A: A short note was added to lines 147-153, explaining why this study is being used and introducing further sections of the manuscript.
What about the COVID-19 pandemic? Have you considered it? How has it influenced the opinion of respondents?
A:In lines 60-65 we have added a section on COVID-19 and the impact on consumer behavior and food waste.
I suggest you to introduce the topic of ugly food and of strategies which can be pursued for reducing food waste also with reference to published case studies of other countries. As your manuscript presents only Polish suggestions, this would be interesting.
A: We have created a new section 1.3 'Strategies to reduce food waste' in which we have included, among other publications suggested by you.
We would like to kindly thank the Reviewer for the time and effort taken to read and review our aricle. We greatly appreciate all the comments and suggestions.
Authors.
Reviewer 2 Report
The work is well done, the topic although it seems " new" is widely covered internationally.
The real significance of these investigations should not be to confirm one's data with the data in the international bibliography, but to create a robust background to operate on the population under investigation with suitable education/awareness programs to impact behaviors also by means of targeted training agendas.
Author Response
Thank you for your positive review. In lines 402-405, the sentence referring to Mr./Mrs. review has been corrected. We would like to kindly thank the Reviewer for the time and effort taken to read and review our aricle. We greatly appreciate all the comments and suggestions.
Authors.
Reviewer 3 Report
The subjectis interesting. The research design is clear and well-structured. The research question is clear. The literature search was developed in depth. From reading section 2.1. I believe there should be articles that used the same delivery model to reinforce your study. In section 2.3. other scientific works that have used the same data analysis methodology should be indicated. The results, comments, and conclusions are well articulated
Author Response
Thank you for your positive review. In sections 2.1 and 2.3, we have added the necessary publications that also use the same survey and statistical methods. We would like to kindly thank the Reviewer for the time and effort taken to read and review our aricle. We greatly appreciate all the comments and suggestions.
Authors.